# Identification of celastrol as a novel HIV-1 latency reversal agent by an image-based screen

**Hongbing Liu[1], Pei-Wen Hu[1], Julien Dubrulle[2], Fabio Stossi[2,3], Bryan C. Nikolai[3], Michael A. Mancini[2,3], Andrew P. Rice[1]***

**1** Department of Molecular Virology and Microbiology, Baylor College of Medicine, Houston, Texas, United States of America, **2** Integrated Microscope Core, Baylor College of Medicine, Houston, Texas, United States of America, **3** Department of Molecular and Cellular Biology, Baylor College of Medicine, Houston, Texas, United States of America

\* arice@bcm.edu

**Data Availability Statement:** Within the manuscript.

**Funding:** This work was supported by the National Institutes of Health grants AI32001 (to A.P.R.).

## Abstract

Although current antiretroviral therapies (ART) are successful in controlling HIV-1 infection, a stable viral reservoir reactivates when ART is discontinued. Consequently, there is a major research effort to develop approaches to disrupt the latent viral reservoir and enhance the immune system's ability to clear HIV-1. A number of small molecules, termed latency reversal agents (LRAs), have been identified which can reactivate latent HIV-1 in cell lines and patients' cells *ex vivo*. However, clinical trials have suggested that combinations of LRAs will be required to efficiently reactivate HIV-1 *in vivo*, especially LRAs that act synergistically by functioning through distinct pathways. To identify novel LRAs, we used an image-based assay to screen a natural compound library for the ability to induce a low level of aggregation of resting primary CD4+ T cells from healthy donors. We identified celastrol as a novel LRA. Celastrol functions synergistically with other classes of LRA to reactivate latent HIV-1 in a Jurkat cell line, suggesting a novel mechanism in its LRA activity. Additionally, celastrol does not appear to activate resting CD4+ T cells at levels at which it can reactivate latent HIV-1. Celastrol appears to represent a novel class of LRAs and it therefore can serve as a lead compound for LRA development.

## Introduction

Although current antiretroviral therapies (ART) are successful in suppressing HIV-1 replication in most individuals, cessation of ART results in reemergence of HIV-1 from a latent viral reservoir, thereby requiring a lifetime ART regime [1,2]. The best described viral reservoir is that of long-lived memory CD4+ T lymphocytes which contain a transcriptionally silent but replication-competent virus integrated in the human genome [3]. Infected macrophages may also serve as a reservoir of latent virus, although the significance of the macrophage reservoir is uncertain [4–6]. Despite its effectiveness, multiple toxicities are associated with ART, especially in the aging population. Long-term ART is associated with osteoporosis, renal and metabolic diseases, and HIV-associated neurocognitive deficits [7–9]. Because of these issues with ART,

Imaging for this project was supported by the Integrated Microscopy Core at Baylor College of Medicine and the Center for Advanced Microscopy and Image Informatics (CAMII) with funding from NIH (DK56338, CA125123, ES030285), and CPRIT (RP150578, RP170719), the Dan L. Duncan Comprehensive Cancer Center, and the John S. Dunn Gulf Coast Consortium for Chemical Genomics.

**Competing interests:** The authors have declared that no competing interests exist.

there is a major effort to develop approaches to disrupt the latent viral reservoir and allow the immune system to clear HIV-1. It is hoped that this strategy, termed "shock and kill," can lead to a functional cure of infection. The "shock" involves the development of strategies to selectively stimulate cells that harbor a latent virus so that RNA Polymerase II (RNAP II) is activated to transcribe the integrated virus, leading to the production of viral antigens that can be recognized by the immune system. The "kill" involves the immune system's ability to recognize and clear cells that express viral antigens.

A number of small molecules have been identified that function as latency reversal agents (LRAs) in CD4$^+$ T cell lines and patient cells *ex vivo* [reviewed in [10–12]]. One class of LRAs are PKC agonists which function through activation of NF-$\kappa$B and the subsequent stimulation of RNAP II transcription via the NF-$\kappa$B sites in the viral long terminal repeat (LTR) sequences. These PKC agonists include prostratin, bryostatin and ingenol derivatives [13–16]. Another class of LRAs are histone deacetylase inhibitors (HDACis) which act by blocking deacetylation of histones, thereby relieving a repressive chromatin state for the viral LTR. HDACi's shown to reactivate latent HIV-1 include vorinostat, panobinostat, and romidepsin [17–19]. JQ1 is an additional LRA that functions by binding to BRD4 and thereby activating CDK9/Cyclin T1 in the P-TEFb complex so that the kinase can be targeted by the HIV-1 Tat protein to activate RNAP II elongation [20–22]. The PKC agonist prostratin and the HDACi's vorinostat and panobinostat also have activities that stimulate CDK9/Cyclin T1 [23,24].

Clinical trials, which have evaluated the ability of a single LRA to reactivate latent HIV-1 in patients, have been disappointing, as no single LRA has been able to reduce the size of the latent reservoir *in* vivo [reviewed in [10,12]]. These failures have led to the notion that combinations of LRAs are required to efficiently reactivate HIV-1 *in vivo*, especially LRAs that function through distinct pathways which synergize to activate latent virus [10,25,26]. Therefore, the identification of LRAs that function through novel pathways is an important research goal. An important property of any LRA is the absence of significant T cell activation, an activity that may induce systemic inflammation.

In an effort to identify new LRAs, we used an image-based assay for compounds that minimally activate primary CD4$^+$ T cells isolated from healthy donors. We used this assay to screen a natural compound library and identified celastrol, a pentacyclic terpenoid, as a novel LRA. Celastrol functions synergistically with bryostatin, HDACi's, and JQ1 in latency reactivation, suggesting a novel mechanism in its LRA activity. Additionally, celastrol does not appear to activate resting CD4$^+$ T cells and it therefore can serve as a lead compound for LRA development.

## Methods

### Cell lines and HIV-1 infections

The Jurkat cell line 2D10 [27] was obtained from Dr. Jonathan Karn and the Jurkat 1G5 [28] was obtained from the NIH AIDS Reagent Program (cat. no. 1819). Cells were cultured in RPMI supplemented with 10% fetal bovine serum (FBS). Jurkat cells were infected with either HIV-1 NL4-3 or HIV-1 NL4-3ΔEnv/Vpr (deleted for *env* and *vpr* genes). Celastrol (0.4 μM) or DMSO control were added at the time of infection. At 17 hours post-infection, cultures were washed with fresh medium and incubated with celastrol (0.4 μM) or DMSO for 24 hours; cell extracts were then prepared for immunoblot analyses.

### Antibodies

Antisera used are: CD25 (catalog no. 12-0259-42, eBioscience), CD69 (catalog no. 11-0699-41, eBioscience), PARP (Catalog no. 95425, Cell Signaling), Cyclin T1 (catalog no. sc-10750, Santa

Cruz Biotechnology), β-actin (catalog no. sa135600204; Sigma), phosphoNF-κB p65, (Ser536, catalog no. 3033S, Cell Signaling), pCDK9 (Thre186, catalog no. 2549, Cell Signaling), Hsp90 (catalog. no. sc-69703, Santa Cruz Biotechnology), NF-κB p65 (catalog. no. sc-109, Santa Cruz, Biotechnology), CDK9 (catalog no. sc-484, Santa Cruz Biotechnology).

## Chemicals

Chemicals used are JQ1 (synthesized by Dr. Damian Young, Baylor College of Medicine), Vorinostat (SAHA; Selleckchem, catalog no. MK0683)), Romidepsin (FK228; Selleckchem, catalog no. S3020), Bryostatin (Sigma, catalog no. 83314), Z-VAD-FMK (Selleckchem, catalog no. S7023).

## CD4 T cell isolation

Resting CD4$^+$ T cells were isolated from healthy donors (Gulf Coast Regional Blood Center, Houston, TX) using the RosetteSep human CD4$^+$T cell enrichment cocktail [STEMCELL technologies, catalogue number (cat. no.) 15062]; activated cells were removed using CD30 Microbeads (Miltenyi Biotec, cat. no. 130-051-401).

## Natural compound library

The natural compound library was purchased from Selleckchem (catalog no. L1400) and contained 173 natural products.

## Image-based screen

6 x 10$^6$ CD4$^+$/CD30$^-$ cells purified from healthy blood donors were resuspended in 20 ml RPMI 1640 medium with 10% FBS and IL-2 at a final concentration of 30 U/ml. Cells were seeded onto duplicate 384-well glass bottom culture plates (Greiner Bio-One, catalog. no. 789836) in which the natural compound library had been distributed in duplicate at a final concentration of 10 μM per compound. DMSO was distributed in duplicate wells in each plate used as a negative control; PMA (Sigma, catalog. no. 781856) plus ionomycin (Sigma, catalog. no. I3909) were used together as positive control in duplicate wells at final concentrations of 10 ng/ml and 1 μM, respectively. The 384-well plates were incubated at 37˚C for two days. Following the incubation, the culture plates were spun at 1,000 rpm for one minute before fixation. Cells were fixed for 30 minutes at room temperature by adding an 8% paraformaldehyde (EMS) solution in PBS directly to each well to prevent cell detachment. After a brief PBS rinse, cells were incubated with a DAPI solution (1μg/ml) for 10 minutes followed by a final PBS wash.

Image acquisition was performed on an IC200 high-throughput microscope (Vala Sciences) using a Nikon PlanApo 10X/0.45 objective. DAPI signal was acquired in 6 image fields per well, and a cell aggregation index was quantified for each well using a custom-made MATLAB script. The aggregation index was defined for each well as the number of DAPI positive pixels from cell aggregates divided by the total number of DAPI positive pixels. The DAPI signal was first segmented after local background subtraction by intensity thresholding using the Otsu method, and the total number of DAPI positive pixels was counted. The resulting objects were then further filtered by size, and objects smaller than 20 pixels (corresponding to the size of 2 to 3 T cells) and larger than 5,000 pixels (most likely autofluorescent non-specific particles) were removed, and the number of DAPI positive pixels from the resulting cell aggregates was determined for the well and divided to the total number of DAPI pixels to obtain the aggregation index. Compounds with an aggregation index greater than 0.08 in at least 3 of the 4

replicates were selected as potential hits, and cell aggregation was manually confirmed by visual inspection of the images.

## Results

### Identification of celastrol as an LRA in an image-based screen

Activation of resting CD4[+]T cells results in cellular aggregation through induction of surface molecules such as LFA1 and ICAM1 [29]. We reasoned that compounds with LRA activity might minimally activate resting CD4[+] T cells and this could result in cellular aggregation that could be detected by automated microscopy. We therefore used aggregation of primary CD4[+] T cells as the readout in an image-based screen of a natural compound library consisting of 173 compounds. To account for potential confounding effects of donor variability, we used resting CD4[+] cells from two healthy blood donors in the screen. Cells were incubated in 384-well dishes for 24 hours in the presence of 10 μM of natural compounds, with DMSO as a solvent control and PMA + ionomycin as a positive control. After the 24-hour incubation, cells were cytospun onto the wells, fixed, and analyzed by automated microscopy (Fig 1A). Quantitation of aggregation indicated that the natural compound celastrol induced a limited amount of aggregation of CD4[+] T cells from both donors (Fig 1B). Celastrol is a pentacyclic terpenoid isolated from the plant *Tripterygium wilfordii* (Fig 1C). A number of biological activities have been associated with celastrol, including anti-oxidant, anti-tumor, anti-inflammatory, and anti-obesity activities [30–33]. Additionally, celastrol has been reported to reduce HIV-1 Tat induced inflammation in astrocytes [34] and inhibit Tat stimulation of RNAP II transcription of the HIV-1 genome [35].

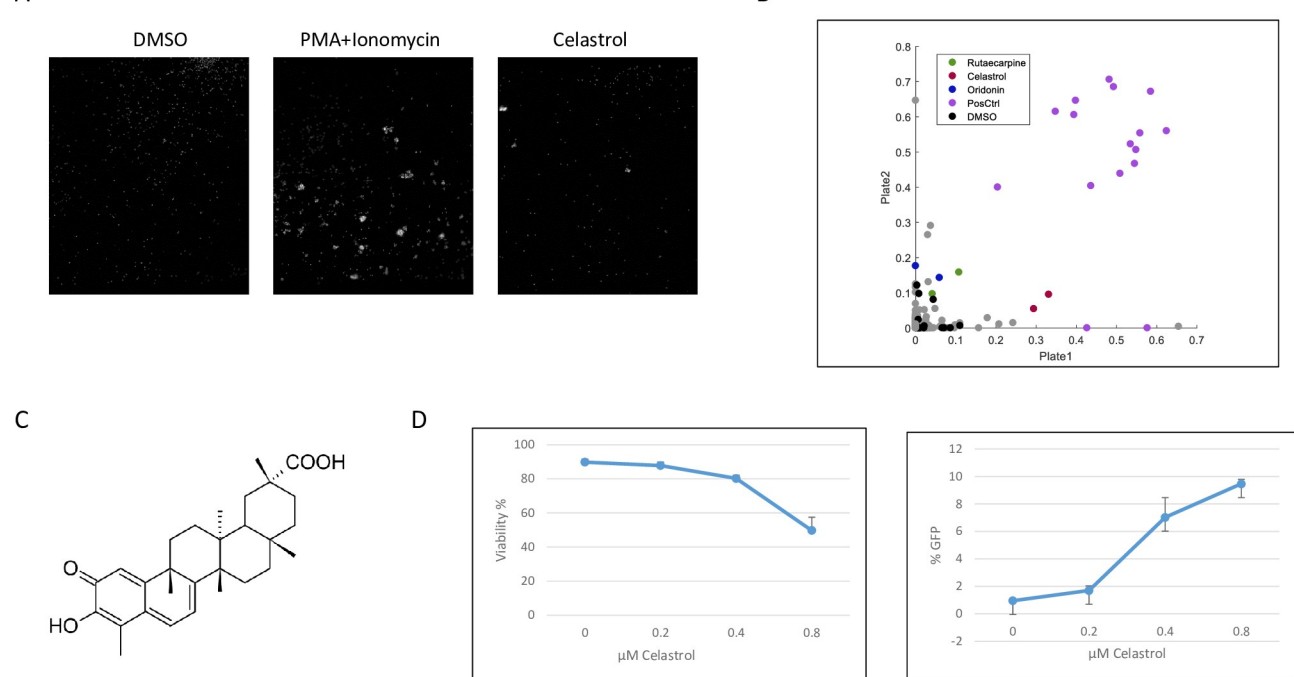

**Fig 1. Image-based screen of natural compound library.** A. Images of resting CD4[+] T cells treated with PMA + ionomycin (positive control), DMSO (negative control), and celastrol (10 μM). B. Scatter plot showing the aggregation index of each compound duplicates in the 2 replicate plates. Positive control replicates are in purple, DMSO negative control replicates in black, inactive compounds in grey and potential hits, namely rutsecarpine, celastrol and oridonin in green, red and blue, respectively. C. Structure of celastrol. D. Jurkat 2D10 cells were treated with the indicated concentrations of celastrol for 24 hours. Reactivation of HIV-1 was quantified by flow cytometry analysis of eGFP expression (left panel); cell viability was quantified by flow cytometry using Vi-Cell (Beckman Coulter). The data presented in Fig 1D are representative of more than three biological replicate experiments.

To determine if celastrol can function as an LRA, the Jurkat 2D10 CD4$^+$ T cell line was treated with a range of celastrol concentrations. This cell line contains a latent HIV-1 virus that expresses an eGFP reporter protein that allows quantitation of viral reactivation. We observed that 0.4 µM and 0.8 µM celastrol reactivated virus in approximately 7% and 10% of cells, (Fig 1D). Cytotoxicity was evident at 0.8 µM celastrol and resulted in loss of viability in 50% of cells. These data indicate that our image-based screen did identify a natural compound, celastrol, with HIV-1 latency reactivation activity in the Jurkat 2D10 cell line.

## Celastrol synergizes with other LRAs

The known properties of celastrol do not provide reliable clues as to what pathway is involved in its LRA activity. We therefore investigated whether celastrol could function additively or synergistically with four well-characterized LRAs: JQ1, vorinostat, romidepsin, and bryostatin. JQ1 activates P-TEFb, vorinostat and romidepsin are HDACi's, and bryostatin is a PKC agonist. An additive activity for celastrol and an LRA would indicate that the two compounds act through the same pathway, while a synergistic activity would indicate that the compounds act through distinct pathways.

We treated 2D10 cells with 400 nM of celastrol and sub-optimal concentrations of the four LRAs and quantified viral reactivation at 24 hours post-treatment (Fig 2). Although 400 nM celastrol had limited reactivation activity by itself, it enhanced the reactivation activity of each of the four other LRAs. To determine whether celastrol displayed synergistic activity with the LRAs, we calculated the Bliss synergy scores [36] for 0.4 µM celastrol plus JQ1 (0.2 µM), Bryostatin (0.2 ng/ml), Vorinostat (0.4 µM), or Romidepsin (20 nM); these Bliss score were 13.2, 8.4, 19.9, and 13.8, respectively, indicating synergism between celastrol and each of these LRAs.

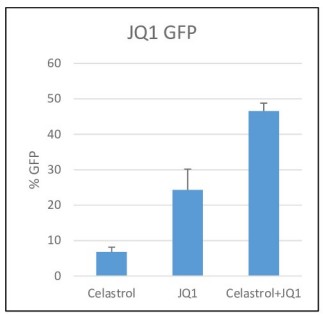 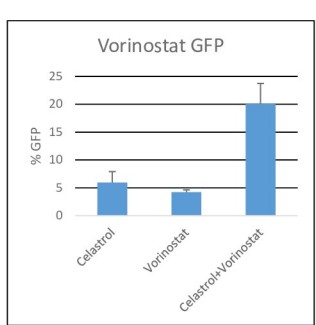 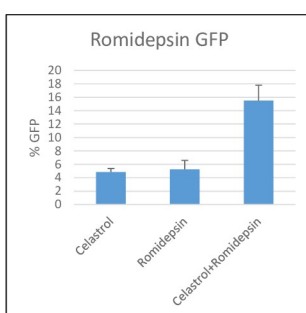 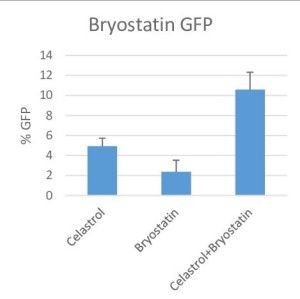

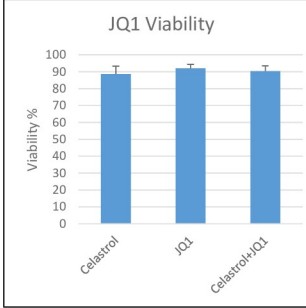 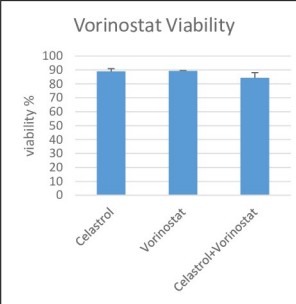 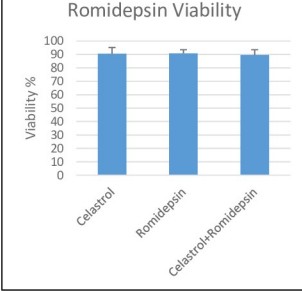 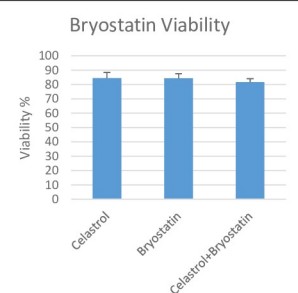

**Fig 2. Reactivation of latent HIV-1 with combinations of celastrol and well-characterized LRAs.** Jurkat 2D10 cells were treated with 0.4 µM celastrol plus 0.2 µM JQ1, 400 nM vorinostat, 20 nM romidepsin, or 0.2 ng/ml bryostatin. Reactivation of latent HIV-1 was quantified by flow cytometry analysis of eGFP expression at 24 hours post-treatment. Cell viability was quantified by Vi-Cell (Beckman Coulter). Bliss synergy scores for 0.4µM celastrol plus JQ1 (0.2 µM), Bryostatin (0.2 ng/ml), Vorinostat (0.4 µM), or Romidepsin (20 nM) were 13.2, 8.4, 19.9, and 13.8, respectively. The data presented in Fig 2 are representative of more than three biological replicate experiments.

These data suggest that the mechanisms of HIV-1 latency reactivation by celastrol is distinct from that of JQ1 vorinostat, romidepsin and bryostatin. Celastrol may therefore represent a novel class of LRA.

## Celastrol reactivation of latent HIV-1 is Tat independent

We utilized the 1G5 Jurkat cell line to determine if the HIV-1 Tat protein is required for the LRA activity of celastrol. This cell line has an integrated HIV-1 LTR that drives expression of firefly Luciferase and does not contain the Tat protein [28]. IG5 cells were treated with either DMSO (solvent control) or 400 nM celastrol plus a range of concentrations of vorinostat, romidepsin, JQ1, or bryostatin. Luciferase expression was quantified at 24 hours post-treatment (Fig 3). Celastrol alone induced only modest levels of Luciferase expression, but it displayed a synergistic effect when used in combination of with each of the LRAs. The data presented in Fig 3 indicate that the LRA activity of celastrol does not require Tat. Additionally, these data further support the conclusion that celastrol may represent a novel class of LRA as it displays synergistic activity with each of these well-characterized LRAs.

## Celastrol does not activate resting CD4$^+$ T cells

Celastrol was identified in our image-based screen as a compound that caused limited aggregation of resting CD4$^+$ T cells (Fig 1). We examined the activation markers CD25 and CD69 to determine whether celastrol induced significant amounts of T cell activation. Resting CD4$^+$ T cells isolated from three donors were treated with 400 nM celastrol, 200 nM JQ1, 400 nM vorinostat, 20 nM romidepsin, and 0.2 ng/ml bryostatin. We evaluated 400 nM celastrol as this concentration has little cytotoxicity and it synergizes with other classes of LRAs. CD25 and CD69 levels were examined by flow cytometry at 24 hours post-treatment (Fig 4). Celastrol did

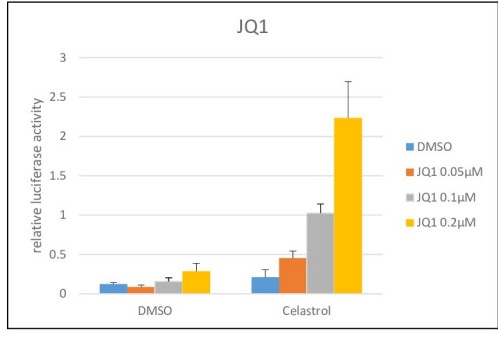
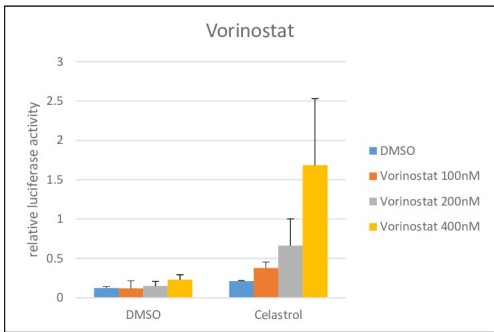
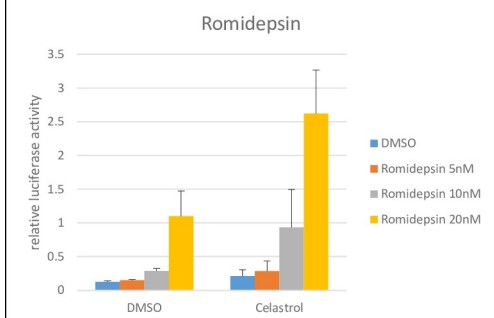
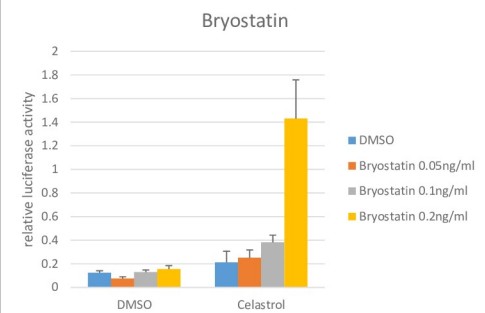

**Fig 3. Reactivation of latent HIV-1 by combination of celastrol and LRAs in Jurkat 1G5 T cell line.** 1G5 cells were treated with DMSO or 400 nM celastrol plus the indicated concentrations of vorinostat, romidepsin, JQ1, or bryostatin. Cell extracts were prepared at 24 hours post-treatment and Luciferase expression was normalized to amount of protein in cell extracts. Data are from three biological replicate experiments.

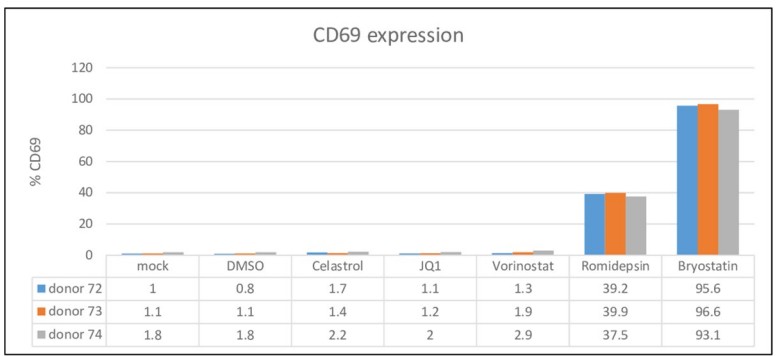

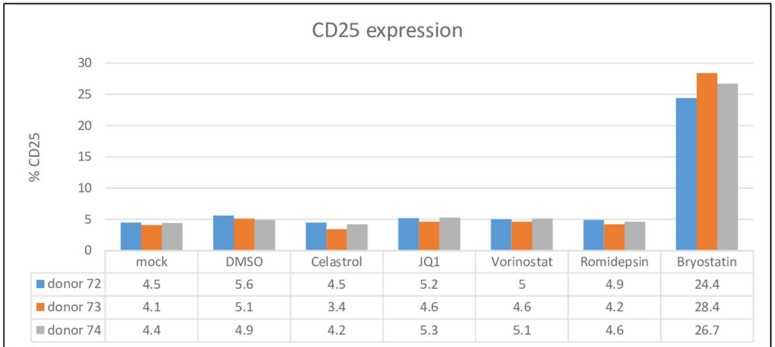

**Fig 4. Celastrol does not activate resting CD4+ T cells.** Resting CD4⁺ T cells were isolated from three healthy donors and treated with 400 nM celastrol, 200 nM JQ1, 400 nM vorinostat, 20 nM romidepsin, or 0.2 ng/ml bryostatin for 24 hours. Expression of CD25 and CD69 was quantified by flow cytometry.

not induce CD25 or CD69 in resting cells from the three donors. CD25 was induced by bryostatin, while CD69 was induced by romidepsin and bryostatin. These data suggest that celastrol has limited effects on T cell activation, despite its identification in our image-based screen by induction of resting CD4⁺ T cell aggregation. However, we used 10 μM celastrol in our screen, a concentration that may have effects on cellular aggregation.

### Celastrol LRA activity does not require apoptosis

Celastrol has been reported to induce apoptosis in several transformed cell lines [37]. We used cleavage of PARP1 as an assay to determine if celastrol induced apoptosis in 2D10 and primary CD4⁺ T cells (Fig 5). Concentrations of celastrol up to 1 μM had little effect on PARP1 cleavage, while 2 μM induced cleavage of the majority of PARP1 in Jurkat 2D10 cells (Fig 5A). In primary CD4⁺ T cells, 50 nM celastrol concentrations induced low levels of PARP1 cleavage and 800 nM celastrol caused total PARP1 cleavage in donor 63 (Fig 5B). Purified CD4⁺ cells from three additional donors were treated with 400 nm celastrol or 200 nm JQ1 and PARP1 cleavage was examined at 24 hours post-treatment (Fig 5C). Modest PARP1 cleavage was observed in cells from donor 72, while only low levels were observed in cells from the other two donors. These data indicate that celastrol induces apoptosis in both Jurkat and primary CD4⁺ T cells, albeit at relatively high concentrations.

Interestingly, Cyclin T1, a component of the Tat co-factor P-TEFb, was strongly down-regulated at 1 μM and 2 μM of celastrol in Jurkat 2D10 cells (Fig 5A). Cyclin T1 contains a PEST sequence at its carboxyl terminus which may be targeted for proteasome-mediated proteolysis [38]. Celastrol has also been shown to inhibit NF-κB activation through inhibition of IκB

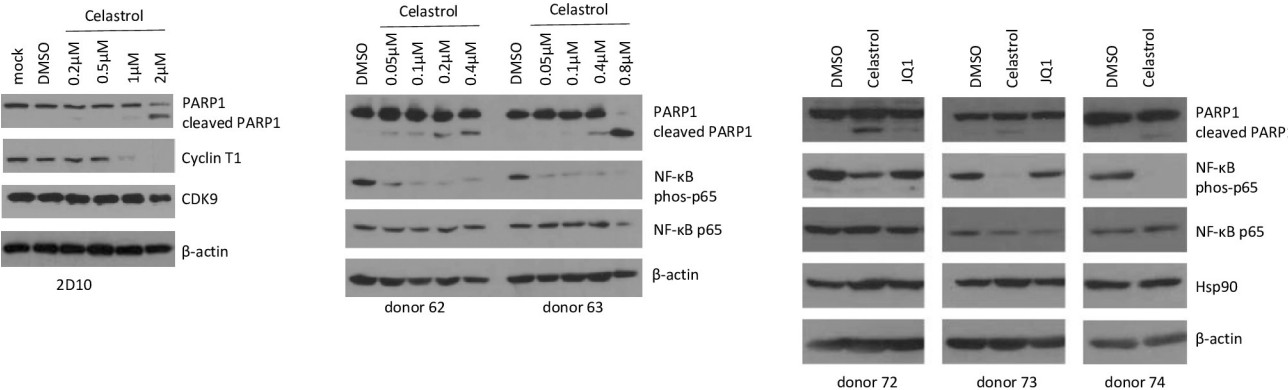

**Fig 5. Induction of apoptosis and inhibition of NF-κB by celastrol.** A. Jurkat 2D10 cells were treated with DMSO or the indicated concentration of celastrol; mock-treated cells did not receive any treatment. Cell extracts were prepared at 24 hours post-treatment and the indicated proteins were analyzed in an immunoblot. B. Resting CD4+ T cells from two healthy donors were treated with DMSO or the indicated concentration of celastrol. Cell extracts were prepared at 24 hours post-treatment and the indicated proteins were examined in immunoblots. C. Resting CD4+ T cells from three healthy donors were treated with DMSO, 400 nm celastrol or 200 nM JQ1. Cell extracts were prepared at 24 hours post-treatment and the indicated proteins were examined in immunoblots.

kinase [39,40]. We observed that levels as low as 0.05 μM celastrol did inhibit NF-κB as indicated by loss of p65 Ser536 phosphorylation (Fig 5B and 5C).

Induction of apoptosis has been reported to reactivate latent HIV-1 [37]. To determine if the LRA activity of celastrol involves induction of apoptosis, we used Z-VAD-FMK to inhibit apoptosis in celastrol-treated Jurkat 2D10 cells. Rather than inhibiting latency reactivation, Z-VAD-ZMK increased reactivation by celastrol, suggesting that apoptosis induced by celastrol impairs latency reactivation (Fig 6). We conclude that the LRA activity of celastrol does not require induction of apoptosis.

## Effects of celastrol on HIV-1 infected cells

To examine effects of celastrol on HIV-1 replication, we infected celastrol-treated (0.4 μM) or DMSO-treated Jurkat CD4 T cells with either HIV-1 wild type NL4-3 virus or the NL4-3 virus deleted for the *env* and *vpr* genes. Cell extracts were prepared at two days post-infection and viral and cellular proteins were analyzed in immunoblots. As expected, celastrol enhanced apoptosis as indicated by increased PARP1 cleavage. Cyclin T1 and CDK9 levels were reduced slightly in celastrol-treated cells relative to β-actin. Despite this reduction in cellular factors required by the viral Tat protein to activate RNA Polymerase II elongation of the viral genome,

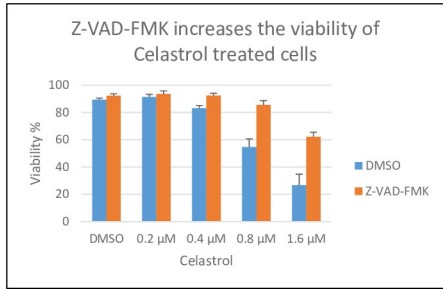
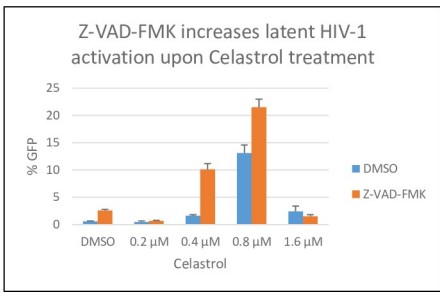

**Fig 6. Inhibition of celastrol-induced apoptosis increases reactivation of latent HIV-1.** Jurkat 2D10 cells were treated with DMSO or the indicated concentration of celastrol and at 24 hours of treatment cell viability was quantified by Vi-Cell (Beckman Coulter) and eGFP expression was quantified by flow cytometry.

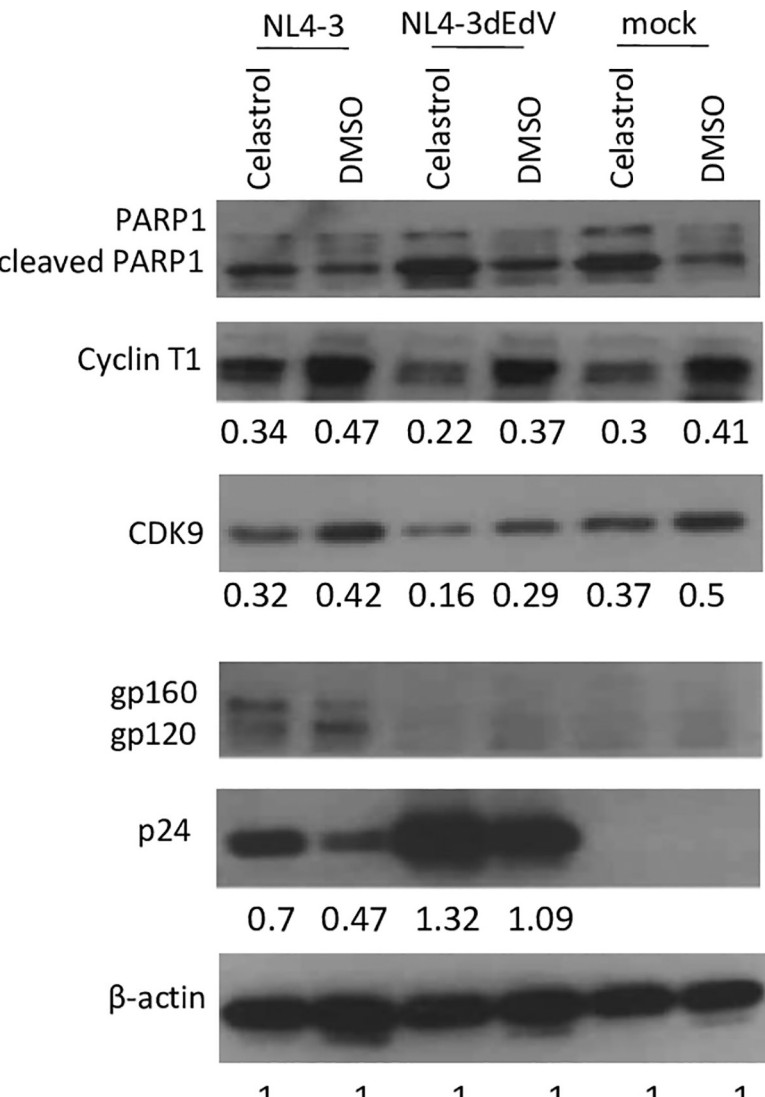

**Fig 7. HIV-1 infection of celastrol-treated cells.** Jurkat cells were treated with 0.4 μM celastrol or DMSO at the time of infection with either wild type HIV-1 NL4-3 or NL4-3ΔEnv/Vpr (deleted for *env* and *vpr* genes). At 17 hours post-infection, cultures were washed with fresh medium and further incubated with celastrol (0.4 μM) or DMSO for 24 hours; cell extracts were then prepared and the indicated proteins were analyzed in an immunoblot. The digital protein images were quantified relative to β-actin by Photoshop. A complete biological replicate experiment as that shown was performed with similar results.

p24 levels were higher in celastrol-treated cells than the DMSO control cells. This is consistent with the finding shown in Fig 3 that celastrol activates viral gene expression by a Tat-independent mechanism. The data presented in Fig 7 demonstrate that celastrol stimulates viral gene expression during productive infections. It is notable in Fig 7 that celastrol appears to inhibit cleavage of gp160 (lanes with NL4-3 virus) by mechanisms that remain to be determined.

## Discussion

Our image-based screen of a natural compound library identified celastrol as a novel LRA when assayed in the Jurkat 2D10 cell line. Our screen utilized aggregation of resting primary

CD4+ T cells as the assay to screen compounds used at final concentrations of 10 μM (Fig 1). However, in our characterization of celastrol's LRA activity, we observed that a concentration of 400 nM celastrol displays LRA activity without induction of the T cell activation markers CD69 and CD25 (Fig 4). At 400 nm concentrations, celastrol also has synergistic reactivation activity with a P-TEFb activator (JQ1), two HDACi's (vorinostat and romidepsin), and a PKC agonist (bryostatin), suggesting that celastrol LRA activity functions through a pathway distinct from these three LRAs (Fig 2). Although induction of apoptosis can reactivate latent HIV-1 [37], celastrol LRA activity is unlikely to involve apoptosis as the low concentrations of celastrol that reactivate latent HIV-1 do not induce apoptosis as measured by PARP1 cleavage (Figs 2, 3 and 5). Additionally, an inhibitor of apoptosis enhances celastrol LRA activity (Fig 6), further suggesting that induction of apoptosis is not involved in this LRA activity. Celastrol LRA activity is Tat-independent as it can activate expression from the HIV-1 LTR in the absence of Tat, as well as synergize with other classes of LRAs in the absence of Tat (Fig 3).

We investigated the LRA activity of celastrol in the CCL19 primary CD4+ T cell model of HIV-1 latency [41]. However, we observed that under the experimental conditions, celastrol displayed cytotoxicity that prevented conclusions about its LRA activity. In the CCL19 protocol, cells are cultured for five days, unlike the T cell activation experiment shown in Fig 4 in which the primary CD4+ T cells were cultured for 24 hours. The CCL19 protocol involves HIV-1 infection and culturing cells in the presence of CCL19. Either the extended time in culture or the HIV-1 infection conditions in the CCL19 model may sensitize primary CD4+ T cells to cytotoxic effects of celastrol. Additionally, celastrol has been reported to inhibit HIV-1 Tat activation of viral gene expression directed by the viral LTR in the U937 cell line [35], a finding that is in contrast to the data presented here. It is possible that different experimental conditions or the use of U937 cells in the previous study may account for these differences. In general agreement with our study is the previous report that celastrol suppressed Tat activation of pro-inflammatory genes through inhibition of NF-κB [34].

Celastrol LRA activity is clearly Tat-independent, as it can activate viral gene expression in the 1G5 cell line that does not express Tat. Interestingly, high levels of celastrol induced the degradation of Cyclin T1 in Jurkat 2D10 cells (Fig 5A). Cyclin T1 contains a PEST sequence at its carboxyl terminus and this sequence may be targeted for proteasome-mediated proteolysis by high levels of celastrol [38]. Induction of NF-κB by celastrol is unlikely to be involved in LRA activity, as celastrol has been reported to inhibit NF-κB [32]. In agreement with this, we observed that celastrol reduces the phosphorylation level of the 65 kDa subunit of NF-κB in Jurkat 2D10 cells (Fig 5). In summary, our image-based screen has identified celastrol as a natural compound that appears to be a new class of LRA and may serve as a lead compound for LRA development. The mechanisms involved in celastrol LRA activity remain to be determined.

## Supporting information

**S1 Raw images.**
(PDF)

## Author Contributions

**Conceptualization:** Julien Dubrulle, Fabio Stossi, Bryan C. Nikolai, Michael A. Mancini, Andrew P. Rice.

**Formal analysis:** Hongbing Liu, Andrew P. Rice.

**Funding acquisition:** Michael A. Mancini, Andrew P. Rice.

**Investigation:** Hongbing Liu, Pei-Wen Hu, Julien Dubrulle.

**Methodology:** Pei-Wen Hu, Julien Dubrulle, Fabio Stossi, Bryan C. Nikolai, Michael A. Mancini.

**Writing – original draft:** Andrew P. Rice.

**Writing – review & editing:** Hongbing Liu, Pei-Wen Hu, Julien Dubrulle, Fabio Stossi, Bryan C. Nikolai, Michael A. Mancini, Andrew P. Rice.

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
