## [Decision Letter · Decision Letter 0]

4 Feb 2021

PONE-D-20-39293

Identification of Celastrol as a novel HIV-1 Latency  Reversal Agent by an Image-Based Screen

PLOS ONE

Dear Dr. Rice,

Thank you for submitting your manuscript to PLOS ONE. After careful consideration, we feel that it has merit but does not fully meet PLOS ONE’s publication criteria as it currently stands. Therefore, we invite you to submit a revised version of the manuscript that addresses the points raised during the review process.

We look forward to receiving your revised manuscript.

Kind regards,

Yuntao Wu

Academic Editor

PLOS ONE

Journal Requirements:

Reviewers' comments:

Reviewer's Responses to Questions

**Comments to the Author**

1. Is the manuscript technically sound, and do the data support the conclusions?

Reviewer #1: Yes

Reviewer #2: Yes

2. Has the statistical analysis been performed appropriately and rigorously? 

Reviewer #1: Yes

Reviewer #2: Yes

3. Have the authors made all data underlying the findings in their manuscript fully available?

Reviewer #1: Yes

Reviewer #2: Yes

4. Is the manuscript presented in an intelligible fashion and written in standard English?

Reviewer #1: Yes

Reviewer #2: Yes

5. Review Comments to the Author

Reviewer #1: In this publication by Liu et al, has found a new class of HIV-1 Latency Reversing Agent (LRA), namely celastrol, following the screening of a natural compound library. The authors very convincingly demonstrated that celastrol quite selectively stimulates latently infected T cells, but not much to the uninfected resting primary CD4+ T cells. However, the precise underlying mechanism through which celastrol works is under investigation. The Rice lab has contributed enormously to the field of HIV transcription and latency, by doing a number of seminal studies. This publication marks the first set of results, which are very exciting and groundbreaking. In this investigation, for disrupting as minimal as possible the cellular state during experimentation, authors used a novel image-based assay.

Overall, this is a very well controlled and organized ongoing study. In this manuscript authors have presented their initial results. Authors convincingly demonstrate the strong synergistic activity of celastrol with other LRAs, however, defining of the underlying molecular mechanism is still under investigation.

Major Concern:

• Given the fact that celastrol synergizes with other LRAs in reactivating latent HIV, it will be interesting to check if celastrol enhances the cell/nuclear translocation of drugs/factors.

• Interestingly, celastrol inhibits NF-kB, and reduces Cyclin T1 levels, but still able to enhance HIV transcription, indeed directing towards a novel mechanism for HIV transcription. I will be crucial to show if celastrol also enhance HIV replication in your system.

Minor Concern:

• Please clearly indicate which NF-kB (p65) residue was assessed for determining the NF-kB phosphorylation.

• Line 75 of page-5 needs editing, maybe they want to say” An important property of any LRA is the absence of significant resting T cell activation, an activity that may induce systemic inflammation”.

• Line 294 of page-14 needs editing, maybe authors want to add inhibition in the sentence “Additionally, celastrol has been reported to INHIBIT HIV-1 Tat activation of viral gene expression directed by the viral LTR in the U937 cell line [35]”.

Reviewer #2: The current manuscript from Rice and colleagues titled “Identification of Celastrol as a novel HIV-1 Latency Reversal Agent by an Image- Based Screen” describes a new and novel compound that may reverse HIV gene expression along with other LRAs. Here the authors used an image-based assay for compounds that minimally activate primary CD4+ T cells isolated from healthy donors and screened a natural compound library and identified celastrol, a pentacyclic terpenoid, as a novel LRA. Celastrol induced a limited amount of aggregation of CD4+ T cells (Low CD25 and CD69 induction) from both donors and functioned additively with JQ1, vorinostat, romidepsin, and bryostatin. Interestingly, Cyclin T1 was strongly down-regulated when adding celastrol in Jurkat 2D10 cells. Overall, the manuscript looks interesting but rather incomplete. It is not clear what is the mechanism of activation for Celastrol, since it decreases phospho-NFKB in both cell lines and donor samples. Also, I am not convinced that the combination drug treatments are not toxic over time. In fact, if the cells are starting to die in 24-48 hrs, that would explain why the virus gets activated and gets out of the cells. Finally, the authors should look for presence of short TAR RNA in their treated samples which can soak up the Tat from the system.

6. PLOS authors have the option to publish the peer review history of their article (what does this mean?). If published, this will include your full peer review and any attached files.

Reviewer #1: No

Reviewer #2: No

---

## [Author Response · Author response to Decision Letter 0]

25 Mar 2021

Response to review

We thank the Reviewer for positive comments and useful critiques that have improve our manuscript. Our response to each concern are:

Major Concern:

• Given the fact that celastrol synergizes with other LRAs in reactivating latent HIV, it will be interesting to check if celastrol enhances the cell/nuclear translocation of drugs/factors.

Response: This is an interesting suggestion and may provide mechanistic insight. However, we believe that the scope of such an analysis is quite broad and would merit its own publication.

• Interestingly, celastrol inhibits NF-kB, and reduces Cyclin T1 levels, but still able to enhance HIV transcription, indeed directing towards a novel mechanism for HIV transcription. I will be crucial to show if celastrol also enhance HIV replication in your system. 

Response: We thank the reviewer for suggesting this experiment and we have performed it. The new data are presented in new Figure 7. Jurkat cells were infected with either HIV-1 NL4-3 or HIV-1 NL4-3Δenv/Vpr and intracellular p24 was quantified in immunoblots. The entire experiment presented in Figure 7 was repeated, so that we have analyzed four independent infections. The data indicate that celastrol modesty enhances p24 expression during infection, despite reductions in Cyclin T1 and CDK9 levels (Tat co-factors). This agrees with data presented in Figure 3 that celastrol functions in a Tat-independent mechanism.

Minor Concern:

• Please clearly indicate which NF-kB (p65) residue was assessed for determining the NF-kB phosphorylation.

Response: We have indicated in the revised manuscript in the Antibodies section of Methods and in the Results (line 252) section that this phosphorylated residue is Ser536. 

• Line 75 of page-5 needs editing, maybe they want to say” An important property of any LRA is the absence of significant resting T cell activation, an activity that may induce systemic inflammation”. 

Response: We have made this suggested edit in the revised manuscript.

• Line 294 of page-14 needs editing, maybe authors want to add inhibition in the sentence “Additionally, celastrol has been reported to INHIBIT HIV-1 Tat activation of viral gene expression directed by the viral LTR in the U937 cell line [35]”. 

• Response: We have made this suggested edit in the revised manuscript.

---

## [Editor Report · Decision Letter 1]

20 Apr 2021

Identification of Celastrol as a novel HIV-1 Latency  Reversal Agent by an Image-Based Screen

PONE-D-20-39293R1

Dear Dr. Rice,

We’re pleased to inform you that your manuscript has been judged scientifically suitable for publication and will be formally accepted for publication once it meets all outstanding technical requirements.

Kind regards,

Yuntao Wu

Academic Editor

PLOS ONE
---

## [Editor Report · Acceptance letter]

21 Apr 2021

PONE-D-20-39293R1 

Identification of Celastrol as a novel HIV-1 Latency Reversal Agent by an Image-Based Screen 

Dear Dr. Rice:

I'm pleased to inform you that your manuscript has been deemed suitable for publication in PLOS ONE. Congratulations! Your manuscript is now with our production department. 

Kind regards, 

on behalf of

Dr. Yuntao Wu 

Academic Editor

PLOS ONE